# Causal relationship between several autoimmune diseases and renal malignancies: A two-sample mendelian randomization study

Puyu Liu[1☯], Jihang Luo[2,3☯], Lanlan Zhao[1], Qingqing Fu[1], Yao Chen[1], Chengfang Li[1], Jieyu Xu[4], Xiaorong Yang[1]*

**1** Department of Clinical Pathology, Affiliated Hospital of Zunyi Medical University, Zunyi, China,
**2** Department of Infectious Diseases, Affiliated Hospital of Zunyi Medical University, Zunyi, China,
**3** Department of Oncology, The First Affiliated Hospital of Jinan University, Jinan University, Guangzhou, China, **4** Department of Pathology, Guiqian International General Hospital, Guiyang, China

☯ These authors contributed equally to this work.
* yangxiaorong2003@126.com

**Data Availability Statement:** All relevant data are available from the GWAS summary database (https://gwas.mrcieu.ac.uk/). The IDs of all data are available in the paper for readers to obtain.

## Abstract

### Objective

Observational studies have shown an association between systemic autoimmune disease (AD) and multiple malignancies. However, due to the difficulty indetermining the temporal nature of the order, their causal relationship remains elusive. Based on pooled data from a large population-wide genome-wide association study (GWAS), this study explores the genetic causality between systemic autoimmune disease and renal malignancy.

### Methods

We took a series of quality control steps from a large-scale genome-wide association study to select single nucleotide polymorphisms (SNPs) associated with systemic autoimmune disease as instrumental variables(IVs) to analyze genetic causality with renal malignancies. Inverse variance weighting (IVW), MR- Egger, weighted median, simple model and weighted model were used for analysis. The results were mainly based on IVW (Random Effects), followed by sensitivity analysis. Inverse-Variance Weighted(IVW) and MR-Egger were used to test for heterogeneity. MR- Egger is also used for pleiotropic testing. A single SNP analysis was used to identify single nucleotide polymorphisms (SNPs) with potential impact. Odds ratio (OR) and 95% confidence interval (CI) were used to evaluate causality, and sensitivity analysis was performed to evaluate pleiotropy and instrumental validity.

### Results

Acute and subacute iridocylitis (*P = 0.006, OR = 1.077*), Ankylosing spondylitis (*P = 0.002, OR = 1.051*), and spondyloarthritis (*P = 0.009, OR = 1.073*) were positively associated with an increased risk of renal malignancy. Coxarthrosis (*P = 0.008, OR = 0.483*), Juvenile rheumatism (P = 0.011, OR = 0.897), and Systemic lupus erythematosus (*P = 0.014, OR =*

**Funding:** The author(s) received no specific funding for this work.

**Competing interests:** The authors have declared that no competing interests exist.

**Abbreviations:** AD, autoimmune disease; SLE, systemic lupus erythematosus; AS, ankylosing spondylitis; JRA, Juvenile rheuma; SpA, Spondyloarthritis; ASIC, acute and subacute iridocyclitis; AAV, ANCA-associated vasculitis; IBD, inflammatory bowel disease; KHOA, knee or hip osteoarthritis; CI, confidence interval; RCT, randomized controlled trial; TNF-α, tumor necrosis factor-α; GWAS, genome-wide association studies; IVs, instrumental variables; IVW, inverse-variance weighted; LD, linkage disequilibrium; MR, Mendelian randomization; OR, odds ratio; SM, simple model; SNPs, single nucleotide polymorphisms; NSAIDs, Nonsteroidal Antiinflammatory Drugs.

*0.869*) were negatively associated with an increased risk of renal malignancy. The results of sensitivity analysis were consistent without heterogeneity or pleiotropy.

## Conclusion

Our study suggests a causal relationship between different systemic autoimmune diseases and renal malignancies. These findings prompt health care providers to take seriously the potential risk of systemic autoimmune disease and provide new insights into the genetics of kidney malignancies.

## Introduction

Systemic autoimmune disease is caused by a variety of factors, such as environmental factors, genetic factors, infection factors, etc. Resulting in extensive deposition of antigen and antibody complex in the blood vessel wall and other reasons leading to damage to multiple organs in the whole body. It is a disease state caused by the wrong immune response of the human immune system to its own components [1, 2]. When the body's own cells or tissue antigens produce an immune response, it is not easy to be completely cleared by the effector cells of the immune system, and it is constantly attacked [3],The organ damage caused by autoimmune diseases and the side effects of long-term use of hormones and immunosuppressive drugs seriously affect the quality of life of patients and bring significant economic burden to patients [4, 5]. Moreover, the association between many autoimmune diseases and cancer is reciprocal. In people with autoimmune diseases, an increased risk of malignant tumors, including solid tumors and non-solid tumors, has been observed, and some malignant tumors also increase the risk of autoimmune diseases [6]. Previous studies on rheumatoid arthritis have shown that cytokines in the occurrence and development of rheumatoid arthritis are related to tumor formation [6]. Patients with systemic lupus erythematosus also have an increased risk of cancer, and age and course of disease are inversely correlated with the risk of cancer [7]. Systemic lupus erythematosus (SLE) may increase the risk of breast cancer and thyroid cancer. In addition, in the study of systemic amyloidosis, it has been found that the most common solid tumor associated with AA amyloidosis is renal clear cell carcinoma, and renal clear cell carcinoma can also induce AA amyloidosis [8]. A study from Swedish data showed that after 26 patients experienced systemic autoimmune disease, not only did the standard incidence rate (SIR) of urinary tumors increase, but overall survival decreased for prostate, kidney, and bladder cancers [9]. An increased risk of RCC and increased mortality in patients with rheumatoid arthritis was also found in southern European populations [10]. Interestingly, other studies have shown the opposite, with a large international cohort study and review showing a decreased incidence of breast and prostate cancer in patients with SLE [11], Recent reports on ANCA-associated vasculitis (AAV) indicate that AAV is associated with the occurrence and development of a variety of cancers, such as lymphoma, leukemia, malignant melanoma, etc. However, another Dutch study believes that the reason is related to the treatment of cyclophosphamide, which has nothing to do with AAV itself [12]. All the above evidence shows that some systemic autoimmune diseases are indeed associated with the occurrence and development of cancer, especially urinary system tumors. However, the mechanism of systemic autoimmune diseases is unknown and related to a variety of external factors, so whether external factors or the disease itself causes the occurrence of malignant tumors is a problem worthy of attention.

Renal cancer is the third largest malignant tumor of the urinary system, and the early symptoms of renal cancer are not obvious, so about 30% of patients have distant metastasis at the first diagnosis [13, 14], and the renal cancer after metastasis is highly resistant to chemotherapy and radiotherapy, so the overall survival of patients is still poor [15]. At present, the mechanism of renal cancer is still unclear. However, a number of studies have shown that kidney diseases are related to autoimmune diseases, lupus nephritis is still the main risk factor leading to death in SLE patients, IgG4 autoimmune diseases usually lead to renal interstitial fibrosis, known as IGG4-associated nephropathy, and the physiological mechanism of the kidney itself is also inextricably linked with the immune system. First of all, the kidney is an essential organ in the human body to maintain immune homeostasis, which is also conducive to clearing inflammatory cytokines and bacterial lipopolysaccharides, and maintaining peripheral tolerance to circulating antigens such as food proteins and hormones. Therefore, through direct and indirect mechanisms, the immune system can contribute to the development of a variety of kidney diseases, acute, chronic kidney disease, and even kidney failure. Whether the incidence of kidney cancer is related to immune system diseases causes us to think, I believe that the relationship between the etiology of kidney cancer and the immune system is worth exploring.

Randomized controlled studies have long been considered the best way to investigate cause and effect. However, due to ethical and other limitations, it is almost impossible to use randomized controlled trial (RCT) in etiological studies, and RCT studies cannot eliminate the confounding effects of environmental factors. Mendelian randomization (MR) is an advanced research design using single nucleotide polymorphisms (SNPs) as instrumental variables (IVs), which has been widely used in etiological studies [16] Because alleles are randomly assigned before the outcome event, the use of SNPs as instrumental variables can reduce the confounding effects of environmental factors. In addition, it can avoid reverse causality bias because genotypes are not affected by disease. In addition, compared with RCT, MR Can use existing open access data from large-scale genome-wide association studies (GWAS), which greatly improves the study scope and statistical power of diseases [17, 18]. Studies have shown that under reasonable quality control and with a large enough sample size, MR Etiology studies based on GWAS data can provide a more reliable level of evidence than RCT [19].

In this study, we used a two-sample Mendelian analysis to evaluate the causal relationship between common systemic autoimmune diseases such as rheumatoid arthritis (RA), systemic lupus erythematosus (SLE), acute and subacute iridocyclitis(ASIC), ankylosing spondylitis (AS), spinal arthritis, hip disease, juvenile rheumatism(JR), systemic connective tissue disease and renal malignancy. This is beneficial to the prevention of renal malignant tumors and further understanding of the relationship between systemic autoimmune disease and malignant tumors.

## Materials and methods

Our aggregated data comes from published studies, all of which have been approved by institutional review boards. The causal relationship between common systemic autoimmune diseases and renal malignancies was verified by two-sample MR.

### Source of database

This study followed the key principles of the STROBE-MR [20] guidelines to ensure methodological transparency and scientific credibility. We utilized the Mendelian randomization study approach to examine the causal association between common systemic autoimmune diseases and renal malignancies. The study methodology is depicted in Fig 1.

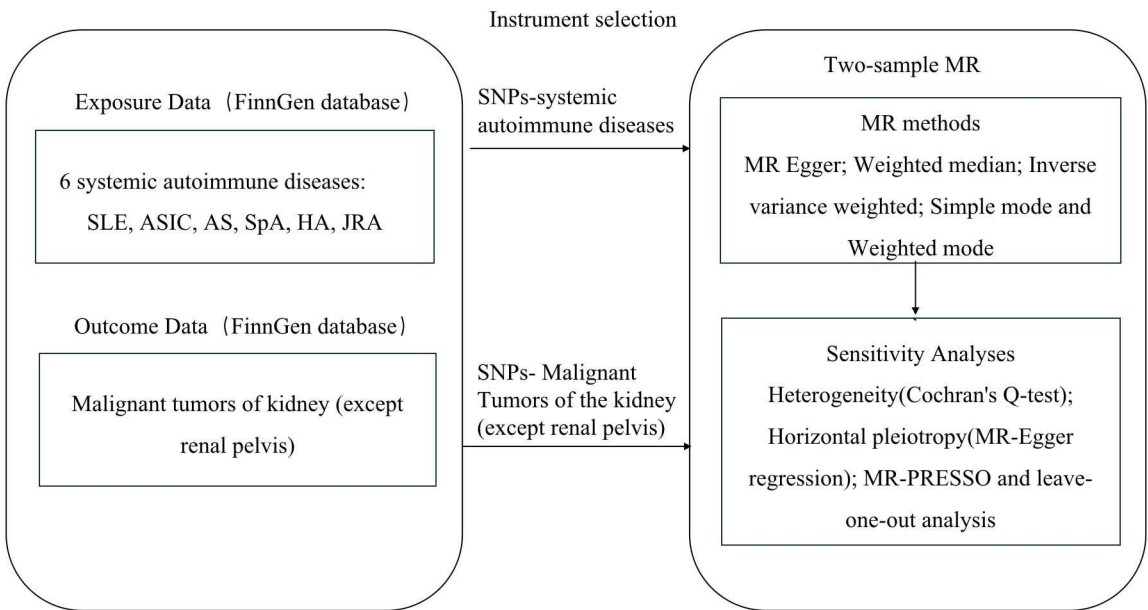

**Fig 1. The research process.** MR, Mendelian randomization; SNP, single-nucleotide polymorphisms; MR-PRESSO, MR Pleiotropy Residual Sum and Outlier.

In order to obtain a more comprehensive causal relationship between systemic autoimmune disease and renal malignancy, we selected genotype data from the FinnGen from the GWAS database.(https://gwas.mrcieu.ac.uk/). Participants were all of European descent (ID: finn-b-H7_IRIDOACUTE acute and subacute iridocyclitis based on 212413 sample size, finn-b-M13_ANKYLOSPON ankylosing spondylitis based on 166144 sample size, finn-b-M13_ARTHTROSIS_COX hip disease based on a sample size of 157930 cases, finn-b-M13_JUVERHEU juvenile rheumatism based on a sample size of 147573 cases, finn-b-M13_SLE based on a sample size of 213683 cases, finn-b-M13_SYSTCONNECT systemic connective tissue disease was based on a sample size of 218,792, and finn-b-SPONDYLOARTHRITIS was based on a sample size of 201,581. The data of their case group and control group are shown in Table 1.

**Table 1. Data sources.**

| Traits | Sample size (cases/controls) | SNPs | Gender | Year | Data sources |
|---|---|---|---|---|---|
| Acute and subacute iridocyclitis (finn-b-H7_IRIDOACUTE) | 212,413 (3,126/209,287) | 16,380,353 | Males and Females | 2021 | FinnGen |
| Ankylosing spondylitis (finn-b-M13_ANKYLOSPON) | 166,144 (1,462/164,682) | 12,243,372 | Males and Females | 2021 | FinnGen |
| Coxarthrosis (finn-b-M13_ARTHTROSIS_COX) | 157,930 (10,709/147,221) | 12,243,487 | Males and Females | 2021 | FinnGen |
| Juvenile rheuma (finn-b-M13_JUVERHEU) | 147573 (352/147,221) | 12,243,324 | Males and Females | 2021 | FinnGen |
| Systemic lupus erythematosus (finn-b-M13_SLE) | 213683 (538/213,145) | 12,243,540 | Males and Females | 2021 | FinnGen |
| Spondyloarthritis (finn-b-SPONDYLOARTHRITIS) | 201,581 (3,037/198,544) | 16,380,410 | Males and Females | 2021 | FinnGen |
| Malignant neoplasm of kidney, except renal pelvis (finn-b-C3_KIDNEY_NOTRENALPELVIS) | 218,792 (971/217,821) | 16,380,403 | Males and Females | 2021 | FinnGen |

Results were also used with FinnGen's renal cancer pooled database, finn-b-C3_KIDNEY_NO-TRENALPELVIS Renal malignancies based on a sample size of 218,792 patients, detailed quality control procedures and association testing methods described in previous publications [21].

## Single nucleotide polymorphism selection

Firstly, SNPs strongly associated with exposure factors ($P<5×10−8$) were screened from the GWAS database. We use a clustering procedure ($r^2 < 0.001$, clustering distance = 10,000 kb) to ensure that there is no linkage imbalance (LD) between SNPs. Third, SNPs with minor allele frequency (MAF) <0.01 were excluded. Finally, after removing the palindromic SNPs, the other SNPs selected are used as IV. To assess the presence of weak bias in the selected IVs, indicating weak associations between the chosen genetic variations as IVs and the exposure factor, we calculated the F-statistic $[F = R2(n−k−1)/k(1−R2)]$. Here, R2 represents the proportion of exposure variance explained by the selected IVs, n is the sample size, and k is the number of IVs. If the F-statistic for the instrument-exposure association is greater than 10, it indicates a low likelihood of weak bias in the instrumental variables [22].

## Statistical analysis

The "TwoSampleMR" R package (version 0.5.7, Stephen Burgess, Chicago, IL, USA) and the "MRPRESSO" package were used for two-sample MR Analysis between exposure and outcome. Two-sample inverse variance weighting (IVW, random effects) was used as the primary analysis method, which consisted of a meta-analysis of SNP-specific Wald ratios between the effect outcome (renal malignance) and exposure (per systemic autoimmune disease) using the random-effects inverse variance method, which weighted each ratio by SE. At the same time, considering the possible heterogeneity in the measurement, MR-Egger, weighted median, simple model(SM)and weighted model were used as supplementary analysis methods. The IVW method can provide an accurate estimate if all the included SNPs can be used as a valid IV hypothesis [23]. R regression can detect and adjust for pleiotrope, but estimates that this method produces very low accuracy [24]. The weighted median gives an accurate estimate based on the assumption that at least 50% of the IV is valid [25].

## Sensitivity analysis

$I^2$ index and Cochran's Q statistics were used for IVW analysis, and Rucker's Q statistics were used for MR-Egger analysis to detect the heterogeneity of SNP effects related to several systemic autoimmune diseases, with $P>0.05$ indicating no heterogeneity [17]. MR-Egger regression is used to identify potential pleotropy and evaluate the impact of pleotropy on intercept test risk estimation. $P> 0.05$ indicates no pleotropy [24]. Leave-one-out analysis is left to identify potentially influential SNPs and evaluate the reliability of the results [26]. Seven types of systemic autoimmune diseases were analyzed in this study. A two-sample MR Study was conducted to assess the causal relationship between systemic autoimmune disease and renal malignancy.

# Results

## IV selection of systemic autoimmune diseases

Detailed information on SNPs associated with systemic autoimmune disease can be found in S1 Table. The SNPs analyzed were all strong IVs, with F values ranging from 29.93 to 1408.80, exceeding the recommended threshold of 10, indicating that there was no bias caused by weak IVs in this study. After screening, a total of 42 SNPs were included for causal analysis of systemic autoimmune disease and renal malignancy (S2 Table).

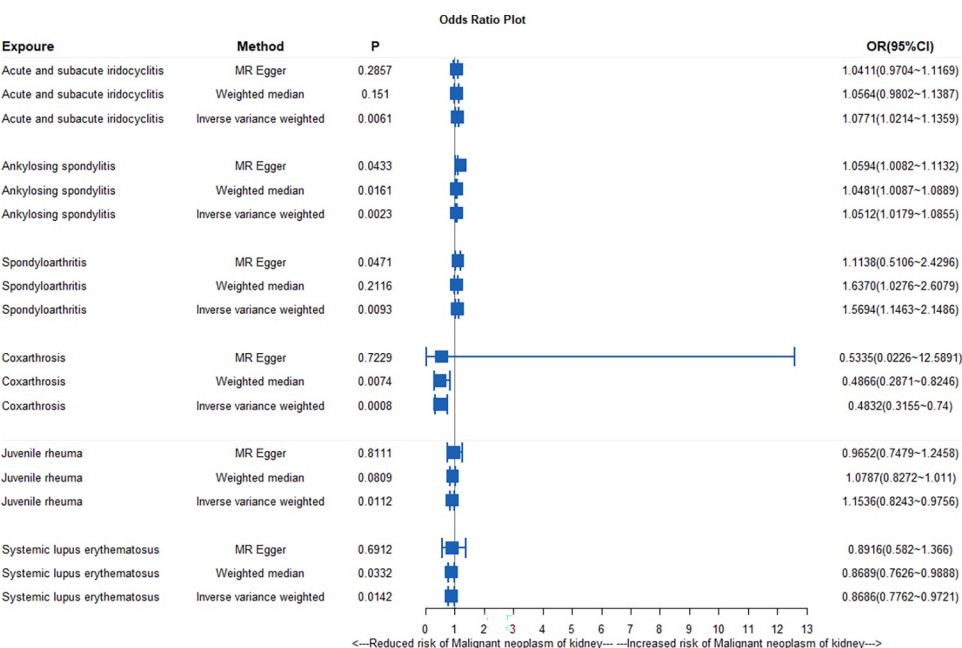

**Fig 2. Mendelian randomization analysis of association between different types of systemic autoimmune disease and malignant tumor of kidney.**

## Mendelian randomization of renal malignancies and three types of systemic autoimmune disease(FinnGen database)

IVW analysis showed that acute and subacute iris mascara (*P = 0.006, OR = 1.077*), ankylosing spondylitis (*P = 0.002, OR = 1.051*) and spinal arthritis (*P = 0.009, OR = 1.073*) were positively correlated with renal malignancy. MR-egger analysis also showed that ankylosing spondylitis and spinal arthritis were positively correlated with renal malignancy (P< 0.05), weighted median analysis showed that ankylosing spondylitis was positively correlated with renal malignancy (*P< 0.05, OR> 1*), and all the results were shown in Fig 2.

Sensitivity analysis based on Cochran's Q test showed no heterogeneity in MR Results between acute and subacute iridocyclitis, ankylosing spondylitis, spinal arthritis and renal malignancy (*P>0.05*). Any association in the MR-Egger regression analysis showed no pleiotropy (*P>0.05*) (S3 Table). The causality of the other two methods (SM and the weighted model) is shown in the S4 Table, and leave-one-out analysis shows that a single SNP does not have a significant effect on the estimated results (S4 Table).

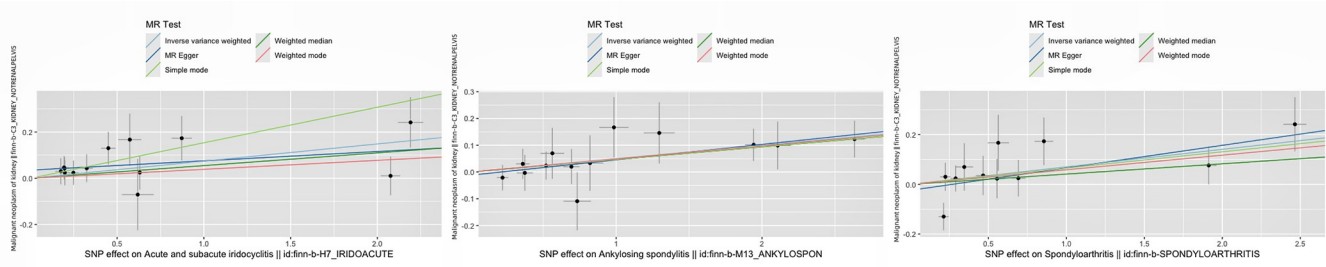

**Fig 3. Scatter plot of effects of single nucleotide polymorphisms on malignant tumor of kidney and three systemic autoimmune diseases (Risk factor).**

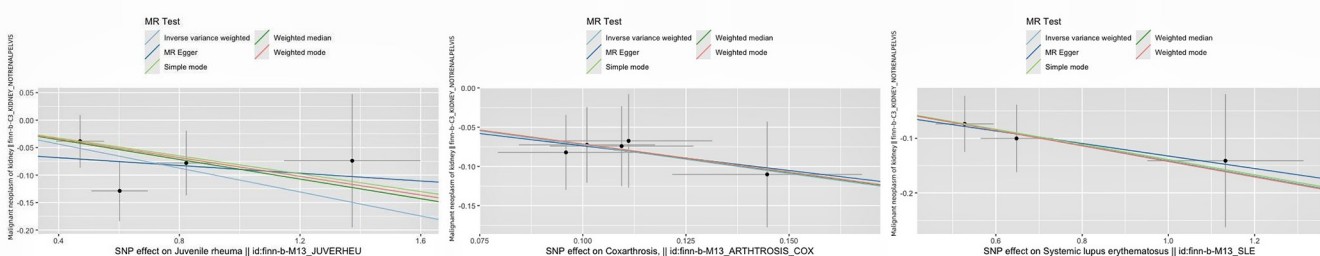

**Fig 4. Scatter plot of effects of single nucleotide polymorphisms on malignant tumor of kidney and three systemic autoimmune diseases (Protective factor).**

The scatter plot in Fig 3 illustrates that three systemic autoimmune diseases are risk factors for renal malignancy. Leave-one analysis showed that no single snp had a significant impact on the estimates (S1 Fig). The funnel and forest plots show SNPs associated with systemic autoimmune disease and their risk of renal malignancy, as shown in S2 and S3 Figs.

### Mendelian randomization of renal malignancies and three other systemic autoimmune diseases (FinnGen database)

IVW analysis showed that hip arthritis (*P = 0.008, OR = 0.483*), juvenile rheumatism (*P = 0.011, OR = 0.897*) and systemic lupus erythematosus (*P = 0.014, OR = 0.869*) were negatively correlated with renal malignancy. Meanwhile, weighted median analysis showed that hip arthritis and systemic lupus erythematosus were negatively correlated with renal malignancy and all the results were shown in Fig 2.

Sensitivity analysis, based on Cochran's q test, showed that we did not find substantial evidence of heterogeneity between hip arthritis, juvenile rheumatism, systemic lupus erythematosus, and renal malignancy *(P > 0.05)*. In addition, MR-Egger regression analysis showed that there was no pleiotropy in this relationship (P > 0.05), as shown in S3 Table. The causality of the other two methods (SM and the weighted model) is shown in the S4 Table, and the "leave one" analysis shows that a single SNP does not have a significant effect on the estimated results (S4 Table).

In addition, Fig 4 visually depicts a scatter plot that illustrates the causal relationship between different systemic autoimmune diseases and renal malignancies.

Finally, the leave-one analysis revealed no significant impact estimates for individual single nucleotide polymorphisms (SNPs) (S1 Fig). S2 and S3 Figs give funnel plots and forest plots showing the relationship between SNPs associated with systemic autoimmune disease and the risk of developing renal malignancy.

## Discussion

The results of this study were selected from the FinnGen Research Project database, which contains all renal malignancies except the renal pelvis, the most prominent type of which is renal clear cell carcinoma, which is diagnosed in nearly 300,000 people worldwide each year and causes more than 100,000 deaths each year The 5-year survival rate for patients with metastatic kidney cancer is 0% to 10% [24]. Current medical methods for systemic treatment of kidney cancer patients do not improve survival in advanced patients. Previous studies on the etiology of renal cancer were mostly based on lifestyle habits and common diseases, such as smoking and obesity, which are important inducers of renal cancer. However, the mechanism is unclear at present [27]. Normal kidneys are affected by various immune diseases, leading to

lupus nephritis, purpura nephritis, nephrotic syndrome, etc. Meanwhile, some patients with immune diseases will use hormone drugs in treatment. This is also another possible factor involving the kidney, but whether kidney malignancy is related to immune diseases, there are no other high-quality, large-scale studies published on this topic.

This study is the first to investigate the genetic causal relationship between different types of systemic autoimmune diseases and the incidence of renal malignant tumors, which makes an important contribution to the study of the mechanism of renal malignant tumors. MR Has great potential for analyzing causal relationships between diseases and traits. In this study, we used a variety of complementary two-sample MR Analysis methods to observe that three systemic autoimmune diseases were risk factors for renal malignancy in individuals of European descent, namely acute and subacute iridocyclitis, ankylosing spondylitis, and spinal arthritis, and four other systemic autoimmune diseases were protective factors for renal malignancy. They are hip arthritis, juvenile rheumatism, systemic lupus erythematosus.

At present, more and more studies have confirmed the close relationship between autoimmune diseases and malignant tumors, but little is known about the role of systemic autoimmune diseases in renal malignant tumors. Systemic autoimmune disease is caused by the extensive deposition of antigen and antibody complex in the blood vessel wall and other reasons leading to multiple organ damage, called systemic autoimmune disease. It is also known as collagen disease or connective tissue disease, which is caused by immune damage leading to celluline-like necrotizing inflammation of blood vessel walls and stroma and subsequent collagen fiber hyperplasia in multiple organs.

Ankylosing spondylitis (AS) is the most common and severe subtype of spondyloarthropathy, and the most common extra-articular manifestation is uveitis, which occurs 20–30% of the time in patients with AS [28]. Acute and subacute iridocyclitis belong to anterior uveitis, which is consistent with the conclusion of this study on acute and subacute iridocyclitis, and the role of AS and iridocyclitis in the etiology of renal carcinoma is consistent. At present, there are few studies on the risk of AS cancer. A recent meta-analysis showed that the overall risk of cancer is increased in patients with AS, especially in Asian populations, where lymphatic and hematopoietic malignancies are strongly associated. The pathogenesis of ankylosing spondylitis, represented by multiple myeloma and lymphoma [29], is related to tumor necrosis factor-$\alpha$(TNF-$\alpha$). TNF-$\alpha$is a cytokine that is involved in the occurrence and development of various cancers through the action of two tumor necrosis factor receptors [30, 31] which may be one of the potential links between ankylosing spondylitis and renal malignancy. As far as we know, no studies have proposed that ankylosing spondylitis is a risk factor for renal malignancy. MR Analysis in this study found that not only ankylosing spondylitis may lead to renal malignancy, but its complication subacute iridocyclitis is also a risk factor for renal malignancy. Moreover, TNF inhibitors, as widely prescribed in patients with AS may also contribute to the observed cancer risk [32], In addition, spinal arthritis also contributes to kidney malignancy in a genetically causal relationship,

Arthritis of the spine is a group of chronic inflammatory rheumatic diseases, including ankylosing spondylitis, reactive arthritis, psoriatic arthritis, arthropathy of inflammatory bowel disease (IBD), etc., this group of rheumatic diseases is also significant in terms of genetic causality for the occurrence of renal malignancies. It has been described previously that ankylosing spondylitis is strongly associated with renal malignancies. A 2020 meta-analysis found that patients with psoriatic arthritis had an increased risk of bladder cancer of 0.31 (*95% CI, 0.19–0.45*), The RR was 1.12 (*95% CI, 1.04–1.19*) [33], suggesting that psoriatic arthritis may have potential implications for urinary system tumorigenesis. IBD is associated with Crohn's disease and ulcerative colitis, and the incidence of IBD is still increasing worldwide in recent years [34].

However, few studies have been conducted on the pathogenesis of IBD, and it has only been proposed that the dysfunctional epithelial barrier may lead to abnormal immune response to intestinal bacteria and eventually cause disease. Recent studies have shown that stromal cells also play an important role in the pathogenesis of IBD in addition to intestinal epithelial cells and inflammatory cells [35]. At present, most studies on IBD around the world focus on treatment. Long-term drug treatment in patients with IBD may lead to nephrotoxicity and nephro-related extratenteral manifestations [36]. This study first proposed that AS, PsA and IBD are strongly associated with the occurrence of renal malignancies. We hope that our MR Study can provide new ideas for the study of immune system disease mechanism of spinal joint.

In contrast, people with systemic lupus erythematosus, hip disease, juvenile rheumatism, and one of the systemic connective tissue diseases also had a reduced risk of kidney malignancy. A 2021 meta-analysis of SLE and 40 malignancies showed that patients with SLE had an increased risk of 24 malignancies, including reproductive cancers, all cancers of the blood system, all cancers of the liver and hepatobiliary system, all cancers of the respiratory system, and gastrointestinal tract cancers, while breast, uterine, melanoma, and prostate malignancies had a reduced risk. For kidney cancer, this study suggests that the risk of malignancy in SLE patients is not affected [29]. The conclusion of this study is different from ours, possibly because single nucleotide polymorphism (SNPs) is a gene-level study of etiology, independent of environmental factors, and the evidence used in meta-analysis is influenced by many factors, such as the quality of the original study, patients' own hormone therapy, lifestyle habits, clinical differences, etc. In addition, we also found that the published meta-analysis studies on SLE did not have exact diagnostic criteria. These studies only used research databases such as the National Center for Primary Health Care Research, the National Health Insurance Claims Database, and the Patient Discharge Dataset, which only recorded the basic daily information of all patients admitted and admitted, such as the date of hospitalization and the date of diagnosis. As well as disease codes, there is no detailed evidence to support the diagnosis of SLE.

Hip arthritis is a chronic disease that leads to a high disability rate in the elderly. It is caused by local joint specific factors working together to increase the load on the whole joint. A study from Sweden showed that the risk of colorectal cancer in patients with hip arthritis may be reduced by 10–20% [37]. Another study from the National Institutes of Health found that compared with the general population, patients with arthritis of the hip or knee have an increased risk of melanoma, renal cell carcinoma, bladder cancer, breast cancer, uterine cancer, and prostate cancer [38]. The mechanisms by which arthritis of the hip is associated with cancer risk are unclear. Some patients with hip arthritis have increased levels of cartilage, bone, or synovial damage in circulating byproducts, but there is no evidence that these are related to the development of cancer. Potential mechanisms that connect knee or hip osteoarthritis (KHOA) and the observed cancer risk may be driven by both the inflammatory processes recently proposed in osteoarthritis and medications used to treat KHOA [39]. A total of 40% to 55% of KHOA patients take non-aspirin Nonsteroidal Antiinflammatory Drugs (NSAIDs) regularly. 10% to 20% of patients take acetaminophen regularly [40, 41].

The use of non-aspirin NSAIDs associated with a reduced risk of colorectal, esophageal, gastric, and colorectal adenomas [42]. In addition, juvenile rheumatism is associated with a reduced risk of renal malignancies. There are differences in the epidemiology and clinical course of rheumatic diseases between adolescents and the elderly [43–45]. The most common rheumatic diseases in adults are rheumatoid arthritis, systemic lupus erythematosus (SLE), and Sjogren's syndrome. The most common are juvenile idiopathic arthritis (JIA), SLE, autoimmune myopathy, and scleroderma [46, 47].

SLE is also more common in adolescents than in adults. The organ damage index of adolescent patients was higher than that of adults. JIA is a concomitant disease of autoimmune disease (AD), may be a precursor or concomitant symptom of another AD, and is also the first cause of chronic arthritis in children. It is suggested that the causal relationship between renal malignancy and rheumatoid disease is related to age and disease course, and the earlier the onset of rheumatoid disease may be a protective factor for renal malignancy. In addition, due to the use of a large number of immunosuppressive drugs in the treatment of patients with systemic autoimmune, some scholars believe that the occurrence of malignant tumors is related to drug use to a certain extent, and azathioprine may increase the risk of lymphoma and other malignant tumors in patients with RA [48–50]. Since this drug is usually used only for patients with severe disease, these findings are likely to be confused with the severe features of the disease itself. It is well known that low doses of glucocorticoids are very effective in controlling inflammatory activity. A study of rheumatoid arthritis veterans showed that glucocorticoid use was a risk factor for non-melanoma skin cancer [51]. However, a large number of studies have shown that glucocorticoid use appears to reduce lymphoma risk [52–55], consistent across disease severity [54]. Oral glucocorticoid use for 2 years had no effect on lymphoma risk, while longer glucocorticoid treatment regimens may reduce lymphoma risk [54]. Therefore, it is still a controversial hypothesis that the use of immunosuppressive drugs affects the occurrence of malignant tumors and needs further study.

We did not find genotypic data for Sjogren's syndrome, dermatomyositis, and Behchet's syndrome in publicly available databases, so the analysis of these three systemic autoimmune diseases could not be conducted in this study. Scleroderma genetic data for local scleroderma were included in the GWAS data, but the screening criteria for strong correlation between exposure factors ($p < 5 \times 10^{-8}$) and clustering process ($r2 < 0.001$, cluster distance = 10,000 kb) could not be included in the analysis.

Our study still has some limitations, first of all, all GWAS data are based on the European population (FinnGen), considering the body size differences among European, North American and Asian populations, our findings are limited to generalizing to other populations, and the conclusions may not cover all ethnicities, but the confidence of Caucasians is still high. There are still some common systemic immune diseases not included in the GWAS database, so a more comprehensive analysis cannot be conducted in this study. Finally, due to the limitations of the method, we cannot rule out that there are still key intermediary factors leading to the occurrence of systemic autoimmune and renal cancer.

## Conclusion

In conclusion, there is a causal relationship between systemic autoimmune disease and increased and decreased risk of renal malignancy. Three systemic autoimmune diseases are risk factors for renal malignancy in individuals of European descent, namely acute and subacute iridocyclitis, ankylosing spondylitis, and spinal arthritis. The other three systemic autoimmune diseases can reduce the probability of kidney malignancy, namely hip arthritis, juvenile rheumatism and systemic lupus erythematosus. Further studies will help to clarify the specific mechanism of kidney malignancy. The results of this study can provide clinicians with predictive help for the possible risk of renal cancer in patients with different autoimmune diseases and reduce the burden on patients.

## Supporting information

**S1 Fig. Leave-one-out of SNPs associated with malignant tumor of kidney and their risk of different types of systemic autoimmune diseases after outliers removal with MR-PRESSO**

**and tightening instrument P value threshold.**
(TIF)

**S2 Fig. Funnel plot of SNPs associated with malignant tumor of kidney and their risk of different types of systemic autoimmune diseases after outliers removal with MR-PRESSO and tightening instrument P value threshold.**
(TIF)

**S3 Fig. Forest plot of SNPs associated with malignant tumor of kidney and their risk of different types of systemic autoimmune diseases after outliers removal with MR-PRESSO and tightening instrument P value threshold.**
(TIF)

**S4 Fig. Scatter plots of the effects of single nucleotide polymorphisms on renal malignancies and rheumatoid arthritis; after removing outliers and tightening the P-value threshold of MR-PRESSO, SNPS associated with renal malignancies and rheumatoid arthritis funnel plot, leave-one-out and risk forest plot were obtained.**
(TIF)

**S1 Table. SNPs associated with different types of systemic autoimmune diseases.**
(XLSX)

**S2 Table. Characteristics of the genetic variants associated with malignant tumor of kidney.**
(XLSX)

**S3 Table. Results of sensitivity analysis.**
(XLSX)

**S4 Table. Mendelian randomization analysis results of simple mode and weighted mode methods.**
(XLSX)

## Acknowledgments

We want to acknowledge the participants and investigators of the FinnGen study. At the same time, we also thank the participants and staff of these open GWAS data.

## Author Contributions

**Conceptualization:** Puyu Liu, Chengfang Li, Xiaorong Yang.

**Funding acquisition:** Chengfang Li.

**Project administration:** Lanlan Zhao, Yao Chen.

**Resources:** Puyu Liu, Jihang Luo, Lanlan Zhao, Qingqing Fu, Jieyu Xu, Xiaorong Yang.

**Software:** Puyu Liu, Lanlan Zhao, Yao Chen, Jieyu Xu.

**Supervision:** Puyu Liu, Jihang Luo, Xiaorong Yang.

**Validation:** Qingqing Fu, Xiaorong Yang.

**Writing – original draft:** Puyu Liu, Jihang Luo, Lanlan Zhao, Xiaorong Yang.

**Writing – review & editing:** Puyu Liu.

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
