## [Decision Letter · Decision Letter 0]

28 Nov 2023

PONE-D-23-31633Causal Relationship Between Common Autoimmune Diseases and Renal Malignancies: A Two-sample Mendelian Randomization StudyPLOS ONE

Dear Dr. Puyu,

Thank you for submitting your manuscript to PLOS ONE. After careful consideration, we feel that it has merit but does not fully meet PLOS ONE’s publication criteria as it currently stands. Therefore, we invite you to submit a revised version of the manuscript that addresses the points raised during the review process.

We look forward to receiving your revised manuscript.

Kind regards,

Kevin Sheng-Kai Ma

Academic Editor

PLOS ONE

Journal Requirements:

Additional Editor Comments:

Thank you for submitting your manuscript. Please make minor revisions according to the reviewers' comments.

Reviewers' comments:

Reviewer's Responses to Questions

**Comments to the Author**

1. Is the manuscript technically sound, and do the data support the conclusions?

Reviewer #1: Partly

Reviewer #2: Yes

2. Has the statistical analysis been performed appropriately and rigorously? 

Reviewer #1: Yes

Reviewer #2: Yes

3. Have the authors made all data underlying the findings in their manuscript fully available?

Reviewer #1: Yes

Reviewer #2: Yes

4. Is the manuscript presented in an intelligible fashion and written in standard English?

Reviewer #1: Yes

Reviewer #2: Yes

5. Review Comments to the Author

Reviewer #1: （1） Autoimmune diseases include many diseases not selected by the authors. For example, why is the more common rheumatoid arthritis not included? It is suggested to adjust the term ‘autoimmune disease’.

（2） Line 49-106 delve into an extensive discussion of the risks associated with autoimmune diseases and cancer. This should be streamlined appropriately as the primary focus of this paper is to introduce the correlation between autoimmune diseases and kidney cancer. The kidneys are the most commonly affected organs in autoimmune diseases. However, it's worth noting that the absence of any citations linking autoimmune diseases to kidney cancer in the text is somewhat inadequate. It is recommended to prominently reference articles on the relationship between autoimmune diseases and kidney cancer in the forefront of the paper.

（3） The quality of images should be enhanced in accordance with the journal's requirements. Lack of figure legend in the manuscript.

（4） In line 285, the discussion mentions that there is no correlation between ankylosing spondylitis and kidney cancer. If there is no correlation, does causality still hold significance? The premise for selecting risk factors and outcome variables is their existing correlation.

（5） In addition to genetic background, whether could the immunosuppressive drugs affect the development of tumors in patients These should be addressed in the Discussing section.

（6） Lack of demographic information, whether different age and sex composition affects the accuracy of the data?

（7） Could the authors provide F-statistics for each individual IV included in MR analysis.

Reviewer #2: This is a great work that comprehensively demonstrated the association between autoimmune diseases and renal malignancies. I only have a few comments:

1. Expand all abbreviations where they were first introduced, not only in abstract but also in the text.

2. "Ankylosing spondylitis (AS) is the most common ...most common extra-articular manifestation is uveitis, which occurs 20-30% of the time in patients with AS" needs a citation. I suggest citing the following reference at the end of this sentence: Ma et al. Management of extra-articular manifestations in spondyloarthritis. Int J Rheum Dis. 2023 Feb;26(2):183-186. doi: 10.1111/1756-185X.14485. PMID: 36703270.

3. Change "NSaids" to "NSAIDs".

4. Line 279: "The pathogenesis of ankylosing spondylitis, represented by ...., but its complication subacute iridocyclitis is also a risk factor for renal malignancy. In addition,..." → TNF inhibitors are widely prescribed to treat AS, which may also interfere with cancer risk. Please change the above sentence to: "The pathogenesis of ankylosing spondylitis, represented by .... , but its complication subacute iridocyclitis is also a risk factor for renal malignancy. Moreover, TNF inhibitors, as widely prescribed in patients with AS [cite the following reference], may also contribute to the observed cancer risk. In addition,..."

[reference] Dong et al. Safety and efficacy of pharmacological treatments for axial spondyloarthritis. Int J Rheum Dis. 2023 Nov;26(11):2130-2133. doi: 10.1111/1756-185X.14853. PMID: 37910029.

5. Line 338: "The only possibility is that the drugs used to treat knee or hip osteoarthritis (KHOA) may be related to the observed cancer risk. A total of 40%..." → Change this sentence to "Potential mechanisms that connect knee or hip osteoarthritis (KHOA) and the observed cancer risk may be driven by both the inflammatory processes recently proposed in osteoarthritis [cite the following reference] and medications used to treat KHOA. A total of 40%..."

[reference] Ma et al. Bidirectional Relationship Between Osteoarthritis and Periodontitis: A Population-Based Cohort Study Over a 15-year Follow-Up. Front Immunol. 2022 Jul 25;13:909783. doi: 10.3389/fimmu.2022.909783

I look forward to reviewing a revised version of the work!

6. PLOS authors have the option to publish the peer review history of their article (what does this mean?). If published, this will include your full peer review and any attached files.

Reviewer #1: No

Reviewer #2: **Yes: **Emily Shu Yen Chan

---

## [Author Response · Author response to Decision Letter 0]

3 Jan 2024

Dear Editors and Reviewers：

Thank you for your letter and for the reviewers’ comments concerning our above-referenced manuscript. Those comments are very valuable for improving our paper. We would like to take this chance to express our appreciation to your time and effort.

We have studied comments carefully and have made some corrections highlighted in the revised manuscript. A point-by-point response has also been prepared and presented in this response letter.

We appreciate for Editorial Office’s and reviewers’ warm work earnestly, and hope the corrections will meet with approval.

Once again, thank you very much for your comments and suggestions.

Yours sincerely,

Puyu Liu

Corresponding author:

Xiaorong Yang

Department of Clinical Pathology, Affiliated Hospital of Zunyi Medical University, Zunyi, China.

Email address: yangxiaorong2003@126.com

Response to reviewers

Reviewer #1

Comment 1: Autoimmune diseases include many diseases not selected by the authors. For example, why is the more common rheumatoid arthritis not included? It is suggested to adjust the term ‘autoimmune disease’.

Reply 1: Thank you for your thoughtful question, which highlights the inadequacy of our approach. We did perform Mendelian analysis on two samples of rheumatoid arthritis and renal malignancy, but the results were not meaningful, as we explained in the discussion section of the article, while also adding data material for rheumatoid arthritis, and adding "several" modifiers before "autoimmune disease" in the title and body.

Changes in the text: 

Title：Causal Relationship Between Several Autoimmune Diseases and Renal Malignancies: A Two-sample Mendelian Randomization Study

Add the 388th to 390th line: Although the genetic data of rheumatoid arthritis(RA) was also included in the GWAS database, we found that there was no causal relationship between rheumatoid arthritis and renal malignancy through Mendelian randomization analysis(Figure S4).

Comment 2: Line 49-106 delve into an extensive discussion of the risks associated with autoimmune diseases and cancer. This should be streamlined appropriately as the primary focus of this paper is to introduce the correlation between autoimmune diseases and kidney cancer. The kidneys are the most commonly affected organs in autoimmune diseases. However, it's worth noting that the absence of any citations linking autoimmune diseases to kidney cancer in the text is somewhat inadequate. It is recommended to prominently reference articles on the relationship between autoimmune diseases and kidney cancer in the forefront of the paper.

Reply 2: We are grateful for your expert appraisal. We take research gaps very seriously, but there are currently few studies focusing on renal malignancies and systemic autoimmune diseases. We quote some from a limited number of articles to supplement the connections between the two. We also simplify the relationship between systemic autoimmune diseases and other cancers.

Changes in the text: 

Delete The 57th to 61th line ；It has been reported that autoimmune diseases are the main cause of death in young and middle-aged women[3], and the organ damage caused by autoimmune diseases and the side effects of long-term use of hormones and immunosuppressive drugs seriously affect the quality of life of patients and bring significant economic burden to patients.

Delete The 67th to 68th line :and it has also been pointed out that people with RA have an increased risk of developing solid tumors.

Delete The 71th to 73th line :Several studies have suggested that SLE is associated with an increased risk of breast and thyroid cancer [9-12].Studies in Taiwan have found that children with SLE have a higher probability of malignant tumor events than children without SLE [13]. 

Delete The 73th to 90th line :Anti-p155/140 antibodies targeting transcriptional mediator 1-γ in polymyositis and dermatomyositis are strongly associated with cancer[14]. Although most studies have shown that systemic autoimmune diseases increase the risk of cancer, there is still some controversy about this conclusion, such as some systemic autoimmune diseases also reduce the risk of cancer or are not associated with the development of cancer.In the study of SLE, a large international cohort study and review showed a decrease in the incidence of breast cancer and prostate cancer[15].Recent reports of ANCA-associated vasculitis (AAV) indicate that AAV is associated with the occurrence and development of a variety of cancers, such as lymphoma, leukemia, malignant melanoma, etc. But another Dutch study linked the cause to treatment with the drug cyclophosphamide, not to AAV itself. The correlation between IGG4-related diseases and cancer is also questionable. Studies have found that patients with IG4RD do not have an increased risk of cancer [16] [17].The above evidence shows that some systemic autoimmune diseases are indeed related to the occurrence and development of cancer.but the mechanism of systemic autoimmune diseases is unknown, and it is related to a variety of external factors, so whether external factors or the disease itself causes the occurrence of malignant tumors is a problem worthy of attention.

Add the 388th to 390th line: Systemic lupus erythematosus (SLE) may increase the risk of breast cancer and thyroid cancer. In addition, in the study of systemic amyloidosis, it has been found that the most common solid tumor associated with AA amyloidosis is renal clear cell carcinoma, and renal clear cell carcinoma can also induce AA amyloidosis[8]. A study from Swedish data showed that after 26 patients experienced systemic autoimmune disease, not only did the standard incidence rate (SIR) of urinary tumors increase, but overall survival decreased for prostate, kidney, and bladder cancers[9].An increased risk of RCC and increased mortality in patients with rheumatoid arthritis was also found in southern European populations.[10] Interestingly, other studies have shown the opposite, with a large international cohort study and review showing a decreased incidence of breast and prostate cancer in patients with SLE [11] ,Recent reports on ANCA-associated vasculitis (AAV) indicate that AAV is associated with the occurrence and development of a variety of cancers, such as lymphoma, leukemia, malignant melanoma, etc. However, another Dutch study believes that the reason is related to the treatment of cyclophosphamide, which has nothing to do with AAV itself [12]。.All the above evidence shows that some systemic autoimmune diseases are indeed associated with the occurrence and development of cancer, especially urinary system tumors. However, the mechanism of systemic autoimmune diseases is unknown and related to a variety of external factors, so whether external factors or the disease itself causes the occurrence of malignant tumors is a problem worthy of attention.

Comment 3: The quality of images should be enhanced in accordance with the journal's requirements. Lack of figure legend in the manuscript.

Reply 3: Thanks for pointing out our problem. We have adjusted all images according to the journal's requirements and placed the figure legend in the manuscript.

Comment 4:In line 285, the discussion mentions that there is no correlation between ankylosing spondylitis and kidney cancer. If there is no correlation, does causality still hold significance? The premise for selecting risk factors and outcome variables is their existing correlation.

Reply 4: We appreciate the insightful question from the reviewer.At present, research on renal malignant tumors and systemic immune diseases is very limited. There are currently no studies that have proposed that ankylosing spondylitis is a risk factor for renal malignant tumors. However, Mendelian randomization studies do not select exposures and outcomes based on correlation. Instead, it uses genotype instrumental variables based on three assumptions: "instrumental variables are strongly associated with exposure factors", "instrumental variables are not associated with confounding factors" and "instrumental variables are only associated with outcomes through exposure". Inferring the relationship between exposure factors and outcomes is fundamentally different from meta-studies.

Comment 5: In addition to genetic background, whether could the immunosuppressive drugs affect the development of tumors in patients These should be addressed in the Discussing section.

Reply 5: Thank you for your suggestions on the completeness of our research. We have added relevant content to the discussion section, which you can see in our relevant paragraphs. Thank you again for your efforts.

Changes in the text: 

Add the 367th to 381th line: In addition, due to the use of a large number of immunosuppressive drugs in the treatment of patients with systemic autoimmune, some scholars believe that the occurrence of malignant tumors is related to drug use to a certain extent, and azathioprine may increase the risk of lymphoma and other malignant tumors in patients with RA [48-50].Since this drug is usually used only for patients with severe disease, these findings are likely to be confused with the severe features of the disease itself. It is well known that low doses of glucocorticoids are very effective in controlling inflammatory activity. A study of rheumatoid arthritis veterans showed that glucocorticoid use was a risk factor for non-melanoma skin cancer [51].However, a large number of studies have shown that glucocorticoid use appears to reduce lymphoma risk [52-55], consistent across disease severity [54]. Oral glucocorticoid use for 2 years had no effect on lymphoma risk, while longer glucocorticoid treatment regimens may reduce lymphoma risk [54].Therefore, it is still a controversial hypothesis that the use of immunosuppressive drugs affects the occurrence of malignant tumors and needs further study.

Comment 6:Lack of demographic information, whether different age and sex composition affects the accuracy of the data?

Reply6：We sincerely thank you for your question. Due to the limitation of the database, we cannot obtain all relevant clinical information. However, because snp widely exists in human variation, it can be used as a good instrumental variable to conduct mr Analysis and get results.

Comment 7:Could the authors provide F-statistics for each individual IV included in MR analysis

Reply 7：Thank you very much for your careful reading of our research content, we have put the data into the supplementary table S3, you can refer to it.

Reviewer #2: 

Comment 1: Expand all abbreviations where they were first introduced, not only in abstract but also in the text.

Reply1: We greatly appreciate the reviewer's positive assessment of our study. 

We have made changes in the manuscript, which you may refer to.

Comment 2:"Ankylosing spondylitis (AS) is the most common ...most common extra-articular manifestation is uveitis, which occurs 20-30% of the time in patients with AS" needs a citation. I suggest citing the following reference at the end of this sentence: Ma et al. Management of extra-articular manifestations in spondyloarthritis. Int J Rheum Dis. 2023 Feb;26(2):183-186. doi: 10.1111/1756-185X.14485. PMID: 36703270.

Reply 2:Thank you very much for your guidance on our manuscript, and we agree with you very much, so we made revisions in the manuscript.

Add cite: [28].Ma KS, Lee YH, Lin CJ, Shih PC, Wei JC. Management of extra-articular manifestations in spondyloarthritis. International journal of rheumatic diseases. 2023;26(2):183-6. Epub 2023/01/28. doi: 10.1111/1756-185x.14485. PubMed PMID: 36703270.

Comment 3:Change "NSaids" to "NSAIDs"

Reply 3:Thank you for your suggestions regarding the shortcomings of our research. We have made changes in the manuscript, which you can refer to.

Comment 4:Line 279: "The pathogenesis of ankylosing spondylitis, represented by ...., but its complication subacute iridocyclitis is also a risk factor for renal malignancy. In addition,..." TNF inhibitors are widely prescribed to treat AS, which may also interfere with cancer risk. Please change the above sentence to: "The pathogenesis of ankylosing spondylitis, represented by .... , but its complication subacute iridocyclitis is also a risk factor for renal malignancy. Moreover, TNF inhibitors, as widely prescribed in patients with AS [cite the following reference], may also contribute to the observed cancer risk. In addition,..."[reference] Dong et al. Safety and efficacy of pharmacological treatments for axial spondyloarthritis. Int J Rheum Dis. 2023 Nov;26(11):2130-2133. doi: 10.1111/1756-185X.14853. PMID: 37910029.

Reply 4:Thank you very much for revising the content of our study, we have adjusted the manuscript, you can refer to it.

Add the 293th to 294th line: Moreover, TNF inhibitors, as widely prescribed in patients with AS may also contribute to the observed cancer risk[32]，

Add cite: [32].Dong C, Braun J, Ma KS. Safety and efficacy of pharmacological treatments for axial spondyloarthritis. International journal of rheumatic diseases. 2023;26(11):2130-3. Epub 2023/11/01. doi: 10.1111/1756-185x.14853. PubMed PMID: 37910029.

Comment 5:"The only possibility is that the drugs used to treat knee or hip osteoarthritis (KHOA) may be related to the observed cancer risk. A total of 40%..." Change this sentence to "Potential mechanisms that connect knee or hip osteoarthritis (KHOA) and the observed cancer risk may be driven by both the inflammatory processes recently proposed in osteoarthritis [cite the following reference] and medications used to treat KHOA. A total of 40%..."[reference] Ma et al. Bidirectional Relationship Between Osteoarthritis and Periodontitis: A Population-Based Cohort Study Over a 15-year Follow-Up. Front Immunol. 2022 Jul 25;13:909783. doi: 10.3389/fimmu.2022.909783

Reply 5:Thank you for your suggestions on the completeness of our research. We have made changes in the manuscript, which you can refer to.

Delete The 348h to 349th line :The only possibility is that the drugs used to treat knee or hip osteoarthritis(KHOA) may be related to the observed cancer risk.

Add The 347th to 349th line :Potential mechanisms that connect knee or hip osteoarthritis (KHOA) and the observed cancer risk may be driven by both the inflammatory processes recently proposed in osteoarthritis and medications used to treat KHOA. [39] 

Add cite: [39].Ma KS, Lai JN, Thota E, Yip HT, Chin NC, Wei JC, et al. Bidirectional Relationship Between Osteoarthritis and Periodontitis: A Population-Based Cohort Study Over a 15-year Follow-Up. Frontiers in immunology. 2022;13:909783. Epub 2022/08/13. doi: 10.3389/fimmu.2022.909783. PubMed PMID: 35958545; PubMed Central PMCID: PMCPMC9358960.

We really appreciate your efforts in reviewing our manuscript during this unprecedented and challenging time. We wish good health to you, your family, and community. Your careful review has helped to make our study clearer and more comprehensive

---

## [Editor Report · Decision Letter 1]

15 Jan 2024

Causal Relationship Between Several Autoimmune Diseases and Renal Malignancies: A Two-sample Mendelian Randomization Study

PONE-D-23-31633R1

Dear Dr. Liu

We’re pleased to inform you that your manuscript has been judged scientifically suitable for publication and will be formally accepted for publication once it meets all outstanding technical requirements.

Kind regards,

Kevin Sheng-Kai Ma

Academic Editor

PLOS ONE

---

## [Editor Report · Acceptance letter]

21 Feb 2024

PONE-D-23-31633R1 

PLOS ONE

Dear Dr. Liu, 

I'm pleased to inform you that your manuscript has been deemed suitable for publication in PLOS ONE. Congratulations! Your manuscript is now being handed over to our production team.

Kind regards, 

on behalf of

Dr. Kevin Sheng-Kai Ma 

Academic Editor

PLOS ONE